# Factors Affecting Attendance of and Attitudes towards Artistic Events among Primary School Children

Judit Váradi [1,*] and Gabriella Józsa [2]

1   Department of Woodwind Instruments, Faculty of Music, University of Debrecen, H-4032 Debrecen, Hungary
2   Pedagogical Institute, Károli Gáspár University of the Reformed Church in Hungary,
    2750 Nagykőrös, Hungary; jozsa.gabriella@kre.hu
*   Correspondence: judit.varadi.06@gmail.com

**Abstract:** In this study, the aim is twofold; on the one hand, we investigate extracurricular artistic activities, and on the other hand, regularity of visits to ten different artistic and cultural events, such as theatres, puppet theatres, dance and folk-dance performances, exhibitions, and musical performances for children, among 9–10 year old primary school children (*N* = 974). We are looking for an answer to the question of whether there is a connection between active artistic activity and the attitude towards cultural and artistic events. Additionally, we plan to identify differences in the family backgrounds of individuals who participate in artistic activities compared to those who do not. Furthermore, we will examine the influence of social, cultural, and economic capital on children's cultural consumption and related attitudes. We tested our hypotheses using SPSS; the methods included logistic regression and factor analysis. In addition to highlighting the role of school in equalising cultural capital, the results confirm the powerful effect of generational transfer. When parents have actively engaged in artistic activities, they tend to offer enhanced support for their children's artistic pursuits. Those who engage in certain types of extracurricular artistic activities during their childhood are more likely to exhibit more positive attitudes towards cultural events as adult consumers. However, parents' high educational attainment and favourable financial circumstances are not the sole determinants of openness to art.

**Keywords:** artistic activity; attendance of art events; artistic experience; cultural transfer





## 1. Introduction

One important role of music education, among others, is to help younger generations become emotionally balanced and productive adults who thrive in life. Therefore, art education and music education should be given primary importance in our pedagogical practice (Váradi 2022). The stimulation of cultural consumption and the opportunity to attend art events have constituted a major focus of recent cultural policy in Hungary. The Performing Arts Act, among the general provisions of Chapter I (XCIX 2008), prioritizes the art education of school-age children. It mandates professional artistic ensembles to promote access to works of art for children and young people and contribute to the effectiveness of public education (Váradi 2016).

The National Core Curriculum (NCC) features a novel approach towards the possibility of live music performance:

> "The cooperation between public education and cultural institutions and organisations is essential to support music education, therefore concert pedagogy should be part of music education" (NCC 2012, p. 10785).

The 2020 changes to the National Core Curriculum have further expanded this possibility. The National Core Curriculum acknowledges and aligns with the fundamental principles of Zoltán Kodály's music education approach, which recognizes choral singing as a pivotal and valuable experimental element. As a result, the curriculum offers avenues for

the implementation of extracurricular musical activities, encompassing concert pedagogy, folk dance education, folk choirs, folk singing groups, and chamber ensembles. These opportunities exist within and beyond the curriculum, emphasizing the importance of collaboration between public education, cultural institutions, and organizations for their successful realization (NCC 2020). We consider it important to introduce younger generations to the values of art as every generation must be given the opportunity to discover and experience art.

According to Erikson's (1997) theory of development, personality development is a lifelong process, whereby the individual is shaped by environmental and social phenomena experienced throughout life. According to his eight-stage system, the role of leisure as a space of identity emerges in stage four (6–11 years). In the school environment, children spend a significant amount of their time learning rather than playing, and they are subjected to constant assessment, which could create a sense of inferiority in the event of failure. At this stage of life, sports and artistic activities contribute to healthy personal development while alleviating and compensating for the pressure to perform at school by providing a sense of achievement.

Art education is based on experiences (Váradi 2020b), which cannot only be taught but must be discovered. The rich context of participation in art events and the reality of a genuine venue provide participants with more experience. Cultural competences can be developed through different branches of art. Various forms of artistic expression also contribute to self-expression, but the creation of opportunities depends on the social context.

## 2. Theoretical Framework

The most well-known figure of French sociology, Pierre Bourdieu (1984), distinguishes economic, cultural, and social types of capital. Of these, economic capital indicates the amount of wealth available for investment, while the social positions of individuals are determined by social capital, i.e., the network of interpersonal relationships. In our research, while examining the impact of the family through our survey, cultural capital emerges as the most significant factor, which Bourdieu (1984) further categorizes into three groups. One form is the embodied cultural capital, which means the knowledge, education, and experience acquired throughout our lives. Another form is the objectified cultural capital, which can refer to a painting or a musical instrument. The final form of cultural capital is the institutionalised cultural capital, including certain types of documents and certifications, such as a professional certificate, diploma, or doctoral degree.

In the context of cultural and social capital, the interests and tastes of younger generations are shaped through the cultural transfer that takes place within the family and during the institutionalised socialisation process (Bauer 2019). Accordingly, the development of cultural tastes is most often examined in a social context, assuming that lower social status is associated with a preference for mass or popular culture and higher status with a preference for so-called high culture or artistic values. Related findings suggest that low socio-economic background adversely affects the development of cultural attitudes, with cultural tastes and social background linked in all ages (DiMaggio 1996). The taste that is formed is reflected in later cultural choices based on the results of previous experiences, which is the channel whereby cultural choices are the result of prior experiences. Therefore, the possibility for individuals to learn about artistic values and thus develop their taste is of primary importance.

Despite the fact that researchers recognise the importance of the dynamics identified by Bourdieu and consequently consider the participation in and appreciation of art as indicators of cultural capital, socio-historical circumstances do not allow this conception of cultural capital to be meaningful for all countries and social groups (DiMaggio and Mukhtar 2004). As for education, research has shown that the appreciation of high culture, complex artistic expressions, is positively associated with educational attainment (DiMaggio and Bryson 1995). Further, by examining the impact of cultural capital on participation in art events from an inter- and intra-societal perspective, Kane (2003) found that Europeans

preferred artistic values. Interestingly, those who possessed much cultural capital valued having friends with similar tastes more than others did (Marsden and Swingle 1994).

The relationship between social stratification and cultural consumption has been systematically described by Chan and Goldthorpe (2007a, 2007b) as being characterised by three main theoretical approaches: the homology, individualisation, and omnivore–univore arguments.

According to the homological approach, upper social classes prefer high culture and lower social classes prefer mass culture, and social and cultural stratification are closely aligned (Sági 2010). Thus, it is possible to distinguish between different status groups coherently and clearly, which is in accordance with Bourdieu's (1984) theory.

The second approach relates to individualisation. According to Bourdieu's (1984) theory of habitus, emphasising the role of the family, a pattern of behaviour is formed during socialization that also influences cultural choice. Proponents of the individualisation theory propose that individuals shape their own lifestyles, which are not solely determined by social factors, such as occupation, income, education, and status (Featherstone 1987). On the other hand, other researchers connect the emergence of cultural consumption to demographic characteristics (Beck 2003), thereby shifting the possibility of choice from Bourdieu's habitus position towards freedom.

The final approach, the omnivore–univore model, mainly focuses on cultural consumption and suggests that the deterministic relationship between social stratification and cultural consumption has become obsolete. Due to the expansion of cultural interests of the upper social strata, people tend to choose a wide range of cultural forms, styles, trends, and genres, thereby encompassing a broad spectrum. This omnivore attitude is primarily determined by the wide range of cultural tastes (Sullivan and Katz-Gerro 2007). At the pole opposite to omnivores or voracious consumers of culture (Peterson 2005), we find univores, who can be demographically characterised as members of young age groups.

In another categorisation, Bauer (2019) also distinguishes between vertical and horizontal forms of cultural value transmission with respect to the formation of cultural preferences. In contrast to vertical generational transfer, which implies a long process and includes the artistic values of past eras, horizontal transfer is more instantaneous and aligns with current trends and fashions. Based on the analysis conducted by Bihagen and Katz-Gerro (2000), it has been found that over the past two decades, cultural competences have become increasingly relevant to problems of social inequality. This finding suggests that regarding the specific artistic values with which young generations are familiar, their discretionary choices and the orientation of their cultural consumption are all irrelevant in this context.

In terms of different theories, our research focuses on understanding the factors that shape the tastes of younger generations. We aim to examine the extent to which family dynamics limit their engagement with the arts, the prevalence of vertical transfer effects, and the opportunities they have to engage in artistic activities and participate in art events.

*Research Objectives*

The aim of our study is to explore the extracurricular artistic activities of students in the fourth grade (9–10 years old) of elementary school in Hungary and their connections with attending art events. Furthermore, we examine how family background affects the active artistic activities of children.

## 3. Method

### 3.1. Participants

The survey was carried out in the framework of the research project, 'Artistic overview on the situation and opportunities of art education and on curricular and extracurricular art activities and events'. The research sample was selected using a multistage stratified probability sampling procedure. The selection of the schools is representative according to the type of settlement (13 county seats, 11 cities, 6 villages), the type of maintenance (21 school districts, 8 churches, 1 other), and we also took into account the proportion of

elementary art education institutions in each county. Data collection took place during regular class sessions by filling out a paper-and-pencil questionnaire. The questionnaire was filled out only by those students who had a consenting parental statement. The survey was conducted among fourth-grade primary school children in three counties of the Northern Great Plain region of Hungary. A total of 974 fourth-grade pupils (9–10 years old) were included in the sample; 52% of respondents were boys, and 48% girls.

### 3.2. Instruments

The questionnaire contained a total of 27 questions. Its first part asked about pupils' gender, age, school, family structure, and the parents' education and employment. The second part asked about pupils' participation in curricular and extracurricular artistic activities and about their attendance of art programmes and events.

Data collection took place in the autumn of 2019 using online and paper-based questionnaires. The number of the ethical approval is 5/2018. In compliance with data privacy regulation, parents were sent information letters about the survey and its content and signed their consent. Pupils completed the questionnaire anonymously in a classroom setting.

The preference for attending art events was measured using a 4-point Likert scale (1: I strongly dislike attending such events. 2: I usually like to but not always. 3: I like to. 4. I really like to.). The variable structure was then examined using exploratory factor analysis. (Table 1).

**Table 1.** Attitudes towards artistic events.

| Statements | Factors | |
|---|---|---|
| | **Musical Omnivore** | **Other Art Lovers** |
| Own value | 2.02 | 1.88 |
| Variance (%) | 25.27 | 23.49 |
| Cumulative (%) | 25.27 | 48.76 |
| Concert . . . | 0.52 | |
| Opera . . . | 0.46 | |
| Theatre . . . | | 0.75 |
| Puppet theatre . . . | | 0.68 |
| Dance performance . . . | 0.72 | |
| Folk dance performance . . . | 0.70 | |
| Exhibition . . . | | 0.78 |
| Folk music concert . . . | 0.68 | |

Note. The table shows the factor weights for the statements.

The reliability of both the full scale (Cronbach's $\alpha$ = 0.74) and the subscales verified by factor analysis (Cronbach's $\alpha_{music}$ = 0.66; Cronbach's $\alpha_{art}$ = 0.64) is acceptable since the literature shows that values above 0.6 are already considered adequate for attitude scales (Gliner et al. 2017).

## 4. Results

### 4.1. Extracurricular Artistic Activities

The focus of this study is on extracurricular artistic activities, which encompass various factors of artistic education that children engage in outside of the compulsory school curriculum. These activities may take place outside of school and include learning to play an instrument, dance, or paint. These activities were included in the survey in three sections each regarding a branch of art: (1) music, (2) visual arts, and (3) dance. The results show that 18% of pupils are learning to play an instrument. A wide range of instruments appear among the responses, but most pupils play the piano (5.6%), followed by the flute (2.1%), the guitar (2%), and the violin (1.2%). Others play the trumpet, flute, cello, or drums. Of the pupils surveyed, 10.8% sing in a group or choir and 12% attend solfeggio lessons. It should be noted here that there may be some overlap, as pupils could select

multiple options. Instrumental group practices are attended by 2.9% of fourth-grade pupils, instrumental folk music practices by 1.5%, and folk choir by 1.7%.

As for visual arts, drawing activities are attended by 12% of the respondents. The second most frequently attended extracurricular activity (9.3%) is arts and crafts lessons. This is followed by photography (2%) and filmmaking (1%), but pupils also attend pottery, painting, felting, and other classes.

Of the various types of dances, folk dance is the most popular (16.7%) followed by modern dance (6%), ballroom dance (4%), majorette (3.5%), and ballet (3%). However, the responses also include dance activities.

Children typically start participating in music activities, such as arts and crafts, when they enter primary school. Extracurricular music lessons mostly take place twice a week, and arts and crafts sessions are usually once a week. Most children attend dance classes twice a week.

*4.2. Attending Cultural Events*

We monitored the regularity of visits to ten different artistic and cultural events, such as theatres, puppet theatres, dance and folk-dance performances, exhibitions, and musical performances for children.

The frequency of attendance of cultural events shows that the most frequently visited cultural institution is the theatre (9.6%). The explanation for such a high rate of theatre attendance is that it is the best known traditional cultural consumption for the family and educators.

This is followed by puppet theatre (6.6%) and then concerts (5.1%). Almost the same proportion of pupils attend dance performances (3%), folk dance performances (2.3%), and exhibitions (2.9%). The least attended events are opera (0.3%) and folk music concerts (0.8%).

We also asked with whom children attended these cultural events (Figure 1).

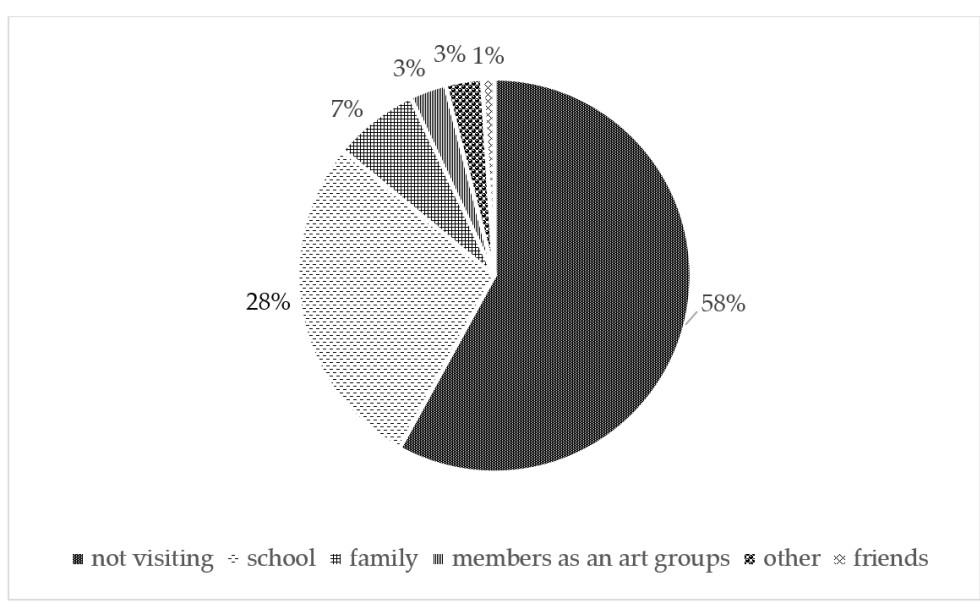

**Figure 1.** Frequency of attendance at cultural events (*N* = 944).

Figure 1 clearly shows that over half of the pupils do not attend any of the listed cultural events. The largest proportion (28%) attends various artistic events organized by their school and class, followed by participation with family and artistic groups, such as choir, orchestra, etc. The school, which is a space for secondary socialisation, contributes significantly to pupils' cultural consumption by organising visits to cultural events.

We created separate aggregate indicators for attendance of music, visual arts, and dance activity. We expected that children who attended one of the extracurricular artistic

activities given earlier were more likely to participate in an event related to that particular branch of art than those who did not engage in artistic activities outside of school.

From going to music, art, and dance classes separately, we created a combined index for each. We are investigating the differences between students who attend extracurricular art classes and those who do not in terms of their frequency of participation in events related to the given art branch. The results of the cross-tabulation analysis are presented in Table 2.

**Table 2.** Cross-tabulation analysis of cultural events and extracurricular activities ($\chi^2$).

| Event | Frequency | Music Activity | | | | Visual Activity | | | | Dance Activity | | | |
|---|---|---|---|---|---|---|---|---|---|---|---|---|---|
| | | No | | Yes | | No | | Yes | | No | | Yes | |
| | | % | *n* | % | *n* | % | *n* | % | *n* | % | *n* | % | *n* |
| Concert | 1 | 2.7 | 18 | 11 | 31 | 4.8 | 36 | 6.3 | 13 | 3.8 | 25 | 8.2 | 24 |
| | 2 | 33 | 222 | 50 | 140 | 35.9 | 268 | 43.2 | 89 | 34.3 | 224 | 45.2 | 133 |
| | 3 | 65 | 439 | 39 | 110 | 59.3 | 443 | 50.5 | 104 | 61.9 | 405 | 46.6 | 137 |
| $\chi^2$ | | 65.43 ** | | | | 5.20 | | | | 22.23 ** | | | |
| Opera | 1 | 0.2 | 1 | 0.7 | 2 | 0.4 | 3 | 0 | 0 | 0.3 | 2 | 0.3 | 1 |
| | 2 | 11.1 | 74 | 14.4 | 40 | 10.5 | 77 | 18.3 | 37 | 12.3 | 79 | 12.2 | 35 |
| | 3 | 88.7 | 590 | 84.9 | 236 | 89.1 | 654 | 81.7 | 165 | 87.4 | 562 | 87.5 | 252 |
| $\chi^2$ | | 4.05 | | | | 9.79 ** | | | | 0.01 | | | |
| Theatre | 1 | 7.4 | 50 | 15.1 | 42 | 8.4 | 62 | 14.9 | 30 | 9.3 | 60 | 10.9 | 32 |
| | 2 | 78.3 | 526 | 76.7 | 214 | 79.4 | 598 | 72.3 | 146 | 76.3 | 493 | 81.9 | 240 |
| | 3 | 14.3 | 96 | 8.2 | 26 | 12.3 | 91 | 12.9 | 26 | 14.4 | 93 | 7.2 | 21 |
| $\chi^2$ | | 17.63 ** | | | | 7.95 * | | | | 10.03 ** | | | |
| Puppet theatre | 1 | 5.4 | 36 | 9.6 | 27 | 6.6 | 49 | 6.9 | 14 | 7.1 | 46 | 5.8 | 17 |
| | 2 | 47.5 | 319 | 58.2 | 168 | 47.6 | 352 | 61.8 | 126 | 44.7 | 289 | 64.4 | 188 |
| | 3 | 47.1 | 316 | 32.1 | 90 | 45.8 | 339 | 31.4 | 64 | 48.2 | 312 | 29.8 | 87 |
| $\chi^2$ | | 20.24 ** | | | | 14.19 ** | | | | 31.97 ** | | | |
| Dance performance | 1 | 1.9 | 13 | 5.7 | 16 | 2.4 | 18 | 5.5 | 11 | 0.9 | 6 | 7.9 | 23 |
| | 2 | 24.3 | 162 | 39.5 | 111 | 26.1 | 193 | 37.8 | 76 | 21.4 | 138 | 44.7 | 130 |
| | 3 | 73.8 | 492 | 54.8 | 154 | 71.5 | 529 | 56.7 | 114 | 77.7 | 501 | 47.7 | 138 |
| $\chi^2$ | | 35.38 ** | | | | 17.39 ** | | | | 96.31 ** | | | |
| Folk-dance performance | 1 | 1.9 | 13 | 3.2 | 9 | 2.3 | 17 | 2.5 | 5 | 0.5 | 3 | 6.5 | 19 |
| | 2 | 35.6 | 238 | 40.8 | 115 | 34.4 | 255 | 47 | 95 | 32.6 | 211 | 47.4 | 138 |
| | 3 | 62.5 | 418 | 56 | 158 | 63.3 | 470 | 50.5 | 102 | 67 | 434 | 46 | 134 |
| $\chi^2$ | | 4.15 | | | | 11.22 ** | | | | 58.01 ** | | | |
| Exhibition | 1 | 2.7 | 18 | 3.5 | 10 | 2.7 | 20 | 3.9 | 8 | 3.6 | 23 | 1.7 | 5 |
| | 2 | 65.8 | 442 | 72.9 | 207 | 66.9 | 497 | 71.4 | 147 | 63.1 | 408 | 78.8 | 234 |
| | 3 | 31.5 | 212 | 23.6 | 67 | 30.4 | 226 | 24.8 | 51 | 33.4 | 216 | 19.5 | 58 |
| $\chi^2$ | | 6.30* | | | | 3.02 | | | | 23.27 ** | | | |
| Folk-music concert | 1 | 0.6 | 4 | 1.4 | 4 | 0.8 | 6 | 1 | 2 | 0.9 | 6 | 0.7 | 2 |
| | 2 | 20.5 | 134 | 34.4 | 96 | 22.7 | 167 | 31.5 | 64 | 19.6 | 126 | 36.1 | 105 |
| | 3 | 78.9 | 526 | 64.2 | 179 | 76.5 | 563 | 67.5 | 137 | 79.5 | 511 | 63.2 | 184 |
| $\chi^2$ | | 22.69 ** | | | | 6.84 * | | | | 29.26 ** | | | |

Note: 1 = every month, 2 = a few times a year, 3 = never. * $p = 0.5$, ** $p < 0.01$.

Table 2 shows that for all extracurricular activity groups, there is at least one cultural event with no difference in the frequency of attendance between those who engage in that artistic activity and those who do not. These events are opera and folk-dance performances for music learners, opera for dance learners, concerts, and contrary to our expectations, exhibitions for visual arts learners. However, we can still conclude that pupils who participate in some kinds of extracurricular artistic activities are significantly more likely to attend cultural and artistic events.

### 4.3. Differences of Family Background in Attitudes towards Art

To characterise the family background, we assessed the educational attainment of both the father and mother, along with an objective index of financial situation and subjective perception of financial situation. We aimed to examine the differences in attitudes towards art along these factors.

The parents' educational attainment was classified into four categories: 1. primary school, 2. vocational training without secondary degree, 3. secondary degree, and 4. university/college degree. Category 5 was "don't know" as not all children aged 9–10 in fourth grade know their parents' highest level of education.

The differences in attitudes towards music and other art by the parents' educational attainment were examined using analysis of variance. The results show that the father's education is associated with significant differences in positive attitudes towards both music ($F = 8.26$, $p < 0.01$) and other art ($F = 14.01$, $p < 0.01$). Tukey's post hoc b test shows that children of fathers with primary education have significantly lower ($M = -0.28$, $SD = 0.74$) attitudes towards other art than children of fathers with higher education ($M = 0.37$–$0.72$, $SD = 1.09$–$1.12$). The picture for the musical omnivore attitude was more varied. Again, children whose fathers have a primary education ($M = -0.12$, $SD = 0.74$) had the lowest attitudes, but the findings are similar for children whose fathers with higher education ($M = 0.18$, $SD = 1.12$). For both music and other art, children whose fathers have a secondary degree have significantly higher attitudes ($M = 0.69$, $SD = 1.12$). The musical omnivore attitude among children whose fathers have a vocational education ($M = 0.35$, $SD = 1.11$) is significantly higher than that among children whose fathers with primary education. However, it is not significantly different from the attitudes of the children whose fathers have a secondary degree and higher education.

There are also significant differences in attitudes towards music ($F = 6.29$, $p < 0.01$) and other art ($F = 13.14$, $p < 0.01$) based on the mother's education. For both attitudes, the same patterns are observed as for fathers' educational attainment, with slightly different means.

The objective financial index was created by adding up the values of the binary variables for each of the following: smartphone, mobile phone, computer/laptop, PlayStation/Xbox, Tablet/iPad, and plasma TV/LCD TV/LED TV/Smart TV. No correlation was found between the objective financial index and the positive attitudes towards music ($r = -0.01$, $p = 0.73$) or other art ($r = 0.02$, $p = 0.64$). Based on these results, we expected no difference according to the subjective financial perceptions either.

The subjective perception of the family's financial situation was examined along four statements (1. We have everything we need and can afford to spend more (e.g., holidays). 2. We have everything we need but cannot afford to spend more. 3. Sometimes we cannot cover our everyday expenses. 4. It often happens that we do not have money to cover our everyday needs). Analysis of variance was used to examine differences in attitudes by subjective financial perceptions. We found no differences in attitudes towards music ($F = 1.84$, $p = 0.14$) or other art ($F = 1.75$, $p = 0.16$) according to subjective material judgements. One reason for this result may be that children in the fourth grade, aged 9–10, are not yet able to fully judge what the family does or does not have money for.

### 4.4. Factors Affecting Attendance of Extracurricular Activities

One of the focal points of this study concerns the factors which influence participation in extracurricular artistic activities. An important determinant is family, including parental support and patterns. In the questionnaire, we asked the pupils about the perceptions of their father' and mother's interests in their academic achievement and artistic activities. The answer is based on the children's opinions, which seems subjective, but at the same time, from their point of view, it is the reality they experience. Pupils could choose from five response options: (1) not at all interested; (2) a little interested; (3) moderately interested; (4) quite interested; (5) very interested. More than half of the respondents (57.5%) feel that their mother is very interested in their academic performance. Slightly

fewer children (42.1%) think this about their father. Only 1.8% think that their mother is not at all interested in their achievement, while 5.1% think the same of their father.

We argue that interest in academic achievement is independent of whether the child attends any extracurricular artistic activities. The results of the two-sample *t*-test (Table 3) for both parents and for all extracurricular artistic activities show no difference between those who attend such classes and those who do not.

**Table 3.** Parents' interests in academic achievements.

| Extracurricular Artistic Activity | Parents | Attend | | Does Not Attend | | *t* (*p*) |
|---|---|---|---|---|---|---|
| | | *M* (*SD*) | *n* | *M* (*SD*) | *n* | |
| Music activity | mother | 4.35 (0.94) | 269 | 4.31 (0.96) | 644 | −0.62 (0.55) |
| | father | 3.93 (1.14) | 260 | 3.92 (1.20) | 616 | −0.04 (0.97) |
| Craft activity | mother | 4.35 (0.92) | 200 | 4.32 (0.96) | 712 | −0.36 (0.72) |
| | father | 4.05 (1.10) | 194 | 3.89 (1.20) | 681 | −1.70 (0.09) |
| Dance activity | mother | 4.32 (0.93) | 289 | 4.33 (0.96) | 624 | 0.10 (0.92) |
| | father | 3.95 (1.15) | 279 | 3.91 (1.20) | 597 | −0.49 (0.62) |

The findings show that 33.2% of mothers and 25.5% of fathers are very interested in their child's artistic activities, whereas 16.3% of mothers and 20.7% of fathers are not at all interested in their child's artistic activities. The latter figures may be biased as pupils who do not attend any extracurricular artistic activities may have indicated parents' lack of interest in such activities.

Unlike academic achievement, for which there is no difference in parental interest between those who attend extracurricular activities and those who do not, a significant difference is revealed in parents' interests in artistic activities between those who attend extracurricular activities and those who do not (Table 4). This result is self-evident since parents cannot be interested in something that their children do not do anyway.

**Table 4.** Parents' interests in (their children's) artistic activities.

| Extracurricular Artistic Activity | Parents | Attend | | Does Not Attend | | *t* (*p*) | *d* |
|---|---|---|---|---|---|---|---|
| | | *M* (*SD*) | *n* | *M* (*SD*) | *n* | | |
| Music activity | mother | 3.92 (1.24) | 254 | 3.24 (1.50) | 476 | −6.21 (0.01) | 0.05 |
| | father | 3.59 (1.36) | 247 | 3.00 (1.47) | 462 | −5.19 (0.01) | 0.04 |
| Craft activity | mother | 3.90 (1.24) | 187 | 3.33 (1.49) | 542 | −4.75 (0.01) | 0.04 |
| | father | 3.61 (1.28) | 184 | 3.06 (1.50) | 524 | −4.49 (0.01) | 0.04 |
| Dance activity | mother | 3.77 (1.29) | 270 | 3.30 (1.51) | 460 | −4.24 (0.01) | 0.03 |
| | father | 3.45 (1.36) | 273 | 3.06 (1.50) | 446 | −3.49(0.01) | 0.03 |

The Cohen's *d* value for the effect size of the difference is small (Leech et al. 2005).

We also asked pupils about their parents' current artistic activities, which 11.6% of mothers and 9.3% of fathers engage in. Since a strong correlation (*r* = 0.53, *p* < 0.01) is observed between the two parents' artistic activities, we combined them to create a new variable for parents' current artistic activities.

Pupils' attendance of extracurricular activities in music (*r* = 0.16, *p* < 0.01), visual arts (*r* = 0.13, *p* < 0.01), and dance (*r* = 0.14, *p* < 0.01) shows a weak but significant correlation with their parents' artistic activities. We are aware from previous research that the family has a strong influence on children's academic performance (Bornstein 2015) and on their further education decisions (Józsa 2022). Our results show that, albeit to a small extent, family background also plays a role in children's attitudes towards art and culture.

The likelihood of pupils choosing to participate in various extracurricular artistic activities was examined using logistic regression. The same variables were included in

the regression model for music, craft, and dance, namely, parents' interests in academic achievements, parents' interests in artistic activities, parents' current artistic activities, children's attitudes towards music, and children's attitudes towards other art. For all three models, backward conditional method was used. The $\chi^2$ test indicates that the regression models are significant ($\chi^2_{music} = 79.36$, $p = 0.01$; $\chi^2_{visual\ arts} = 68.51$, $p = 0.01$; $\chi^2_{dance} = 72.46$, $p = 0.01$). For musical extracurricular activities, the individual effects of all included independent variables are significant other than the variable of parental interest in academic achievement. For craft activities, there is no significant individual effect of parents' interests in academic achievements and pupils' attitudes towards other art. For dance classes, there is no unique effect of parental involvement in art as an independent variable. The $Exp(\beta)$ odds ratio reported in the table shows that the likelihood of attending an extracurricular artistic activity grows as values of the significant independent variables increase.

The results of the logistic regressions (Table 5) show that the likelihood of participating in all three extracurricular activities increases when pupils hold more favourable attitudes towards music and when parents demonstrate greater interest in their children's artistic activities. This highlights the importance of both the family and the school in educating primary school children to become art enthusiasts and cultural consumers during their adolescence and adulthood.

**Table 5.** Regression analysis of extracurricular activities and independent variables.

|  | Independent Variables | β | Wald | *p* | Exp(β) |
|---|---|---|---|---|---|
| Music lessons | Parents' interests in artistic activities | 0.59 | 4.37 | 0.04 | 1.81 |
|  | Parents' current artistic activities | 0.32 | 21.93 | 0.01 | 1.37 |
|  | The children's attitudes towards music | 0.39 | 22.47 | 0.01 | 1.47 |
|  | The children's attitudes towards other art | 0.25 | 8.19 | 0.04 | 1.29 |
| Craft activities | Parents' interests in artistic activities | 0.25 | 12.50 | 0.01 | 1.28 |
|  | The children's attitudes towards music | 0.26 | 10.24 | 0.01 | 1.30 |
| Dance lessons | Parents' interests in artistic activities | 0.15 | 5.81 | 0.02 | 1.16 |
|  | The children's attitudes towards music | 0.56 | 45.47 | 0.01 | 1.75 |
|  | The children's attitude towards other art | 0.21 | 5.94 | 0.02 | 1.24 |

Nagelkerke $R^2_{music} = 15.4\%$; Nagelkerke $R^2_{visual\ arts} = 6.2\%$; Nagelkerke $R^2_{dance} = 14\%$.

The correlation between attendance of cultural events and attitudes towards music and other art was explored through correlation analysis. We expected that music lovers attended related events at a higher rate than those who appreciated other art (Table 6).

**Table 6.** Correlation of cultural events and attitude factors towards art (r).

| Variables | Concert | Opera | Theatre | Puppet Theatre | Dance Performance | Folk Dance Performance | Exhibition | Folk Music Concert |
|---|---|---|---|---|---|---|---|---|
| Musical omnivore | 0.49 ** | 0.43 ** | 0.14 ** | 0.16 ** | 0.64 ** | 0.62 ** | 0.14 ** | 0.61 ** |
| Other art lover | 0.30 ** | 0.02 | 0.46 ** | 0.52 ** | 0.03 | 0.15 | 0.65 ** | 0.18 ** |

** $p < 0.01$.

As Table 6 shows, attitudes towards branches of art other than music have a strong significant relationship with attendance of theatre, puppet theatre, and exhibitions, which are cultural events unrelated to music. The findings reveal that taking children to the types of cultural events they are interested in and appreciate can further strengthen the foundation for their future cultural consumption.

## 5. Discussion

According to the well-known paradigm from cultural sociology, the cultural capital brought from the family and the school as a secondary socialization arena have a significant influence on children's artistic sensitivity. Parents influence their children's performance in

many ways (Szűcs and Péter 2020). Cultural sociology research has confirmed that the high cultural activities of those with diplomas is more significant, which also affects the cultural capital brought from the family. ter Bogt et al. (2011) found that cultural preferences show continuity from generation to generation.

Encouraging cultural consumption and attendance of art events has become a priority in recent years. To this end, the art education of children and young people in Hungary, including the introduction of artistic works to them, is addressed in both the Performing Arts Act and the National Core Curriculum. In addition to curricular education, the artistic activities of the family and parents also make a significant contribution to pupils' consumption of culture.

Our research involved 974 fourth-grade pupils in Northeastern Hungary. The aim of this study has been to examine the characteristics of fourth graders' participation in cultural events, their extracurricular artistic activities, and the factors that contribute to a higher likelihood of participating in such activities. We considered three groups of extracurricular artistic activities and cultural events: music, craft, and dance. Our results show that less than half of the surveyed pupils attend extracurricular artistic activities. The most popular activity is music, followed by dance and visual arts. Most pupils initiate their chosen activity after starting school and engage in it on a weekly basis.

As for attendance of artistic events, our findings suggest that most often, children attend art performances with their school classes or families. The most frequently attended cultural events (9.6%) are theatre performances, followed by puppet theatres (6.6%), and concerts (5.1%). In terms of attendance of cultural events, the school, which is a space for secondary socialisation, contributes significantly to children's cultural tastes.

Bauer's (2019) research concluded that theatre visits often occur with the family, while other art venues, such as concert halls and museums, are typically visited with school organisations. The role of the school was also found to be significant in the DEXARTA survey, with 70% of pupils having attended different art events together with their school classes (Váradi 2020a). The results of the research suggest that the role of family, parents, grandparents, and relatives at this age is not as significant as that of school.

A specific form of encouraging cultural consumption was investigated by Damen and Van Klaveren (2013), who summarised the experience of cultural and art education in the Netherlands and found that by the end of the programme, participation in events of high culture had increased even though pupils could use their cultural vouchers for any event of their choice.

The institutionalised cultural and economic capital of pupils' parents does not necessarily determine attitudes towards artistic events. According to the theory proposed by Chan and Goldthorpe (2007a), the omnivore model, also referred to as culturally greedy model by Peterson, is primarily determined by a wide range of cultural tastes (Sullivan and Katz-Gerro 2007). This implies an inclination towards an artistic polyglotism, resulting from the broadened horizons of cultural consumption. With respect to attitudes towards artistic events, pupils are divided into two groups: musical omnivores, who have a positive attitude towards all music-related events (concerts, opera, dance, folk dance, folk music) and those who prefer other art (theatres, puppet theatres, exhibitions).

The likelihood of attending extracurricular artistic activities was examined separately by branches of art. The independent variables included in the analysis display varying significance across the examined branches of art. At the same time, children's attitudes towards music and parents' interests in their children's artistic activities are the significant determinants in all three cases.

## 6. Limitations

Despite the representative sample of the study, the generalisability of the results is limited as there were limitations to the research. A self-administered questionnaire was used to collect data. On the one hand, there is much literature on the bias of self-report measures (Booth-Kewley et al. 2007; Richman et al. 1999). On the other hand, the responses

are based on the perceptions and opinions of 9–10 year olds, which may bias the results depending on the degree of subjectivity.

In future research, it is worthwhile to extend the data collection to parents' perceptions or other family-related factors to strengthen the validity of the results.

## 7. Conclusions

Hargreaves and colleagues found that children are open to all styles of music until they are 9–10 years old. This phenomenon was called "open-earedness", but after that, tastes change, the influence of peers becomes more and more important, and the values of adults are rejected. At the end of youth, as experiences are reflected on, it has been found that childhood musical experiences contribute to the development of adult tastes (Hargreaves et al. 1995).

The most significant results of our research are recognition of the factors affecting the cultural consumption among 9–10 year old students. Overall, we can say that children who participate at least in one of extracurricular artistic activity are significantly more likely to attend cultural and artistic events.

Our results confirm the significant influence that the family and the school exert on young people's participation in extracurricular activities and highlight the importance of extracurricular artistic activities in cultural consumption.

Those who engage in some kinds of artistic extracurricular activities have the potential to become cultural consumers in adulthood. However, it is important to note that the high educational attainment and favourable financial circumstances of parents do not necessarily determine one's openness to art.

**Author Contributions:** Conceptualization, J.V.; methodology, G.J.; writing—original draft preparation, J.V. and G.J.; writing—review and editing, J.V. and G.J.; project administration, J.V.; funding acquisition, J.V. All authors have read and agreed to the published version of the manuscript.

**Funding:** Research Institute of Art Theory and Methodology of Hungarian Academy of Art.

**Institutional Review Board Statement:** The study was conducted in accordance with the Declaration of Helsinki, and approved by the Institutional Review Board (or Ethics Committee) University of Debrecen (protocol code 5/2018 and date of approval, 25 October 2018).

**Informed Consent Statement:** Informed consent was obtained from all subjects involved in the study. Written informed consent has been obtained from the participants via the school administrators to publish this paper.

**Data Availability Statement:** The data presented in this study are available on request from the corresponding authors. The data are not publicly available due to the survey group consisting of minors; thus, it was the institutions' specific request not to make the data publicly accessible.

**Conflicts of Interest:** The authors declare no conflict of interest. The funders had no role in the design of the study; in the collection, analyses, or interpretation of data; in the writing of the manuscript; or in the decision to publish the results.

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
