# Peer review of "Factors Affecting Attendance of and Attitudes towards Artistic Events among Primary School Children"

_socsci, doi:10.3390/socsci12070404_

Round 1

Reviewer 1 Report (New Reviewer)

The title as well as the introduction raised expectations about your manuscript and research. The topic you are addressing would be a relevant addition to existing literature. Thank you for this valuable contribution. I will structure my feedback in (a) general remarks (these comments cover feedback applicable in the entire manuscript), and (b) specific remarks (feedback on sentence and/or word level). The specific remarks can include a quote from your original manuscript to refer to a specific section. The specific remarks will refer to page (emphasis added in boldface; e.g., 1.15/16) and row(s; e.g., 11.15/16).

General remarks:

The overall manuscript is relevant for existing literature. To increase the quality of the manuscript, you need to pay attention to clarity of writing and consistency in terminology and displays (see below).

Specific remarks:

p.title              You are talking about art events, but in the main text about artistic events. Keep this consistent.

p.1.6                10 = ten.

p.1.8                The “n” needs to be placed in italics.

p.1.11              SPSS program = SPSS (this is terminology used in research for this).

p.1                   Describe that your aim is twofold to guide the reader better.

p.keywords     Remove the 1, 2, etc. This is redundant.

p.abstract        The font size is different? Please check.

p.1.28              To what does the XCIX refer?

p.1.34              You haven’t introduced the abbreviation NCC yet. Place this after the first time you mention the complete word.

p.1.35              “this possibility” = To what does this refer?

p.1.42/43         This is odd. As a reader, I do not know why you place questions here.

p.2                   This page is fragmented. Try to create more coherence via signaling words and connector sentences.

p.2.54              “only discovered” = This seems counterintuitive when it comes to education because you are talking about teaching with learning and a consequence. Learning—as a concept—consists of discovering what works and what does not. This is something different than you are describing.

p.2.55              “more information” = About what?

p.2.59              Mention year of publication.

p.2.62/63         “is the most relevant” = Why?

p.2.67              “certain types of documents and certifications” = Such as?? The latter asks for an example.

p.2.72              The term/ concept “cultural tastes” requires an example.

p.2.78              There is a double space.

p.2.81              “appreciation of high culture” = What does this mean?

p.2.85              “the role of women was higher” = Do you mean the influence? Because the way you describe it is incorrect (i.e., this is probably not how you mean that).

p.2.88              For this paragraph you need a signaling word to connect it to the previous paragraph. In addition, a paragraph is at least three sentences long. As a result, this is not a paragraph.

p.3.96              Insert publication year after the author.

p.3.99              Avoid back-to-back brackts: (x)(y) = (x; y).

p.3.110            You are talking about “young age groups”, but this isn’t only a geographical characteristic?

p.3.117            “not irrelevant” = Relevant. The double negative is a no-go.

p.3.120–123    Why is the section highlighted?

p.3.125            The extracurricular has to be mentioned sooner in the manuscript.

p.3.126            Use an en dash instead of a hyphen.

p.3.127            “looking for looking” = Please revise. Make sure you use an appropriate verb (I would suggest to use “to examine”).

p.3.135            Why do you label it as “type of maintenance”? I would also suggest to present the information is a table.

p.4.151/152     Did parents give explicit consent for your study? Only an information letter is not sufficient.

p.4.Table1       What do the numbers mean in the table? Is it an average score? Moreover, you need to look into reporting on two or three decimal numbers. In addition, I would suggest to use a period instead of a comma to avoid confusion.

p.4.160            The word “music” and “art” need to be placed in subscript.

p.5.182            You use the Oxford comma inconsistently. Go over your manuscript to make this consistent.

p.5.189            10 = ten.

p.Figure1        Use patterns instead of colours.

p.5.202            N needs to be placed in italics.

p.6.217            “some kind of” = Vague.

p.Table2          The N needs to be a small letter as you are talking about a part of your sample (and not your complete sample). The letter also needs to be placed in italics. Furthermore, the alignment in the table is off. Please check. I would also suggest to move the table to the appendices. You also need to review the decimal numbers.

p.7.224            This has to be better explained in the methodology (it is not properly introduced).

p.7.229            What is your argument to cluster secondary degree and technical school?

p.7.232–245    Place the correct letters in italics, such as M, F, and p (and apply this throughout your manuscript). Moreover, if you present the mean score you also need to present the standard deviation (SD).

p.8.276            “not at all” = just “not”. There is no need to use intensifiers.

p.10                 The spacing is different. Please adjust.

p.11.339          “well-known paradigm” = Redundant.

p.11.346/347   Requires a source. Please insert one.

p.11.352          “has been to understand” = To examine.

p.11.357          This paragraph does not follow logically.

p.11.374          “cultural consumption” = Aim for more consistent terms and descriptions.

p.11.381          Double space.

p.12.399          I would suggest to change “parents” to family.

p.12.403          Earlier in your manuscript you were talking about 9–10 year-olds?

p.12.416          “some kind of” = Vague.

p.references    En dash versus hyphen (it should be an en dash) between the page numbers.

See above. 

Author Response

Thank you for your comments, which have helped us improve the study.

Reviewer 2 Report (New Reviewer)

The paper is an interesting account of cultural participation among 9/10 year olds in Hungary. However, the paper addresses two aspects - interest in arts activities and participation - without setting out the reasons for looking at both aspects and how they might be related. Given the cross-sectional nature of the data, the author should be wary of implying that the child's attitudes towards music/other arts influences their participation as the causality could be in the other direction. 

The correlation between parental and child activities is quite low but is taken as an indicator of cultural socialisation. I think the discussion of this could be more nuanced.

I think the suggestion that current patterns of participation mean that children will participate in cultural activities in adulthood is a little strong, given the available evidence. 

The language is mostly fine but would benefit from some final proofing. 

Author Response

Thank you for your comments, which have helped us improve the study and refine our conclusions. The paper has been proofread by a native speaker.

Round 2

Reviewer 1 Report (New Reviewer)

Comma and period use in statistics is still inconsistent (compare table 1 with the paragraph below it). 

Table 2 looks sloppy. The formatting needs attention to make it easier to read. Can you please check and play around with settings?

N and n aren't used properly (N for complete sample; n for sub part of the sample. Please go over your manuscript and revise (e.g., in Table 2). The N/n also need to be placed in italics. Also apply this to the remainder of your manuscript. 

See above. 

Author Response

Point 1. Comma and period use in statistics is still inconsistent (compare table 1 with the paragraph below it). 

 Response 1: Yes, you are totally right, we also fixed the remaining comma error.

Table 2 looks sloppy. The formatting needs attention to make it easier to read. Can you please check and play around with settings?

 Response 2: The table was re-edited based on the APA7 guidelines

N and n aren't used properly (N for complete sample; n for sub part of the sample. Please go over your manuscript and revise (e.g., in Table 2). The N/n also need to be placed in italics. Also apply this to the remainder of your manuscript. 

Response 3: Thank you for the comment, we have corrected it in the manuscript.

Reviewer 2 Report (New Reviewer)

I do not feel that the author has sufficiently addressed my comments on the previous version. The paper addresses two aspects - interest in arts activities and participation - without setting out the reasons for looking at both aspects and how they might be related. This rationale should be included.

Given the cross-sectional nature of the data, the author should be wary of implying that the child's attitudes towards music/other arts influences their participation as the causality could be in the other direction. 

The correlation between parental and child activities is quite low but is taken as an indicator of cultural socialisation. I think the discussion of this could be more nuanced.

Adequate

Author Response

I do not feel that the author has sufficiently addressed my comments on the previous version. The paper addresses two aspects - interest in arts activities and participation - without setting out the reasons for looking at both aspects and how they might be related. This rationale should be included.

Response 1. Thank you for your comments, which have been fully taken into account. We supplemented the question of the research with the purpose for which we also investigated artistic activity and participation in art events. With this research, we also looked for an answer to the question of whether there is a connection between active artistic activity and the attitude towards cultural and artistic events.

Given the cross-sectional nature of the data, the author should be wary of implying that the child's attitudes towards music/other arts influences their participation as the causality could be in the other direction. 

Response 2. Yes, you are absolutely right, but based on our results, this statement was confirmed. But we constantly emphasize that this connection is possible. However, we rejected the hypothesis that this relationship might influence cultural consumption in adulthood.

The correlation between parental and child activities is quite low but is taken as an indicator of cultural socialisation. I think the discussion of this could be more nuanced.

Response 2. Our results were also supported by the results of other researches.

Round 3

Reviewer 2 Report (New Reviewer)

This version has taken on board my previous comments.

Author Response

Response to Reviewers

This is an interesting piece, well researched and ready for publication. There are two minor amendments needed.

Dear Reviewer,

We are grateful for your kind comment and valuable contribution to our study. Thank you for your assistance; it is greatly appreciated.

  1. Check sentence flow and wording in abstract in particular and then throughout the remainder of the article. there are some minor English language corrections needed: the whole piece just needs a proof read to check for correct use of sentence and paragraph construction.

Indeed, you are correct. We have thoroughly reviewed the entire manuscript and made the necessary corrections. Our particular focus was on ensuring the abstract section is accurately revised.

  1. Do quotes that are longer than two lines need to be in separate paragraphs?

Yes, we greatly appreciate your input. We have taken your advice and effectively paraphrased the longer quotations, ensuring they are accurately conveyed within a well-structured paragraph.

Thank you sincerely for your valuable and helpful feedback.

This manuscript is a resubmission of an earlier submission. The following is a list of the peer review reports and author responses from that submission.

Round 1

Reviewer 1 Report

As stated in the abstract the study aims to "investigate extracurricular artistic activities as well as participation in cultural art events among fourth-grade primary school children". The authors have conducted a survey among children aged 9-10 years old (N= 974), and are especially interested in how family background influence those “who do and do not participate in artistic activities”. The results are based on statistical analyses.

Introduction. The main focus in this section is on the legitimization on arts education for young children with the reference to national curriculum documents and personal psychology theory. There are a few references made to literature. The Introduction do not present the necessary background for the study, nor is an explicit purpose stated or explicitly stated research questions.

Materials and methods. In this section some concepts from sociological theory and research on musical taste are referred to, and they may prove relevant to the study, but the RQs are not discussed or presented in this section either. In addition, the theoretical concepts – the different forms of capital – are not defined nor linked explicitly to the survey questionnaire. It also seems as musical taste (of whom – the children and/or the parents?) is not an issue in the survey as far as the reader knows from what is presented from the survey. Thus, the theoretical foundation for the study seems non stringent. There is a lack of information about the survey and also about how the participants were selected.

There is also a major issue with the validity of the data as young children are set to answer questions about the family and their parents as for example, the parents income and their educational status as well about the frequency of attending art events. Another example, is that the child’s experience of whether or not his/her parents are interested in their academic achievement and artistic activities, later on is referred to as “Parents’ interest in academic achievement” and “Parents’ interest in (their child’s) artistic activities” (see for example, Table 2 and Table 3).

That is, the data on the parents background all seems to be based on how the 9-10 year old perceives it. This section also lacks to clarify what is meant by omnivores and univores in this specific setting as the concept of musical genres is not discussed.

Results. The chosen statistical analyses seem relevant, but most of them seem not valid considering the validity of the data on parents' background/interest/taste.  

Author Response

Response to Reviewer 1 Comments

Comments and Suggestions for Authors

Point 1: As stated in the abstract the study aims to "investigate extracurricular artistic activities as well as participation in cultural art events among fourth-grade primary school children". The authors have conducted a survey among children aged 9-10 years old (N= 974), and are especially interested in how family background influence those “who do and do not participate in artistic activities”. The results are based on statistical analyses.

Introduction. The main focus in this section is on the legitimization on arts education for young children with the reference to national curriculum documents and personal psychology theory. There are a few references made to literature. The Introduction do not present the necessary background for the study, nor is an explicit purpose stated or explicitly stated research questions.

 Response 1: Thank you for your suggestions, which have helped us improve the study's quality.
All your comments have been incorporated into our paper.
In the Introduction, we have presented the purpose of the research and the research questions.

Point 2: Materials and methods. In this section some concepts from sociological theory and research on musical taste are referred to, and they may prove relevant to the study, but the RQs are not discussed or presented in this section either. In addition, the theoretical concepts – the different forms of capital – are not defined nor linked explicitly to the survey questionnaire. It also seems as musical taste (of whom – the children and/or the parents?) is not an issue in the survey as far as the reader knows from what is presented from the survey. Thus, the theoretical foundation for the study seems non stringent. There is a lack of information about the survey and also about how the participants were selected.

Response 2: We have presented the research questions. We have defined the forms of capital, focusing on the incorporated form of cultural capital relevant to our study. We introduced the research, the sample and questionarry in more detail.

Point 3: There is also a major issue with the validity of the data as young children are set to answer questions about the family and their parents as for example, the parents income and their educational status as well about the frequency of attending art events. Another example, is that the child’s experience of whether or not his/her parents are interested in their academic achievement and artistic activities, later on is referred to as “Parents’ interest in academic achievement” and “Parents’ interest in (their child’s) artistic activities” (see for example, Table 2 and Table 3).

Response 3: Yes, we know, that the answer is based on the children's opinion, which seems subjective, but at the same time, from their point of view, it is the reality they experience and essential for us too.

Point 4: That is, the data on the parents background all seems to be based on how the 9-10 year old perceives it. This section also lacks to clarify what is meant by omnivores and univores in this specific setting as the concept of musical genres is not discussed.

Response 4: The concept of omnivores was used in the sense that, of the different events presented and studied, these children preferred all events that involved any type of music (concert, opera, folk music, and dance performance).

Point 5: Results. The chosen statistical analyses seem relevant, but most of them seem not valid considering the validity of the data on parents' background/interest/taste. 

Response 5: From the children point of view, the data are valid because they show the reality of children.

Reviewer 2 Report

Interesting and important article that highlights the importance of arts in a school environment and that early exposure in a child's life is central to a person continuing to participate and consume artforms as an adult. It is also interesting to read about Hungary's art education in schools and how it relates to the cultural heritage that Hungary has in the Zoltán Kodály method and other pedagogies.

Some improvements for suggestions:

In abstract: define "art" , is it artforms you mean?

In text:

2. Define what you mean by "high and low culture" in a contemporary context. Bourdieu is not enough, needs to be further analysed with relevant updated references. 

Describe the parent group, how was the questionnaire distributed? 

Write the age of the pupils in the study, not only the grade. 

3. Define extracurricula artistic activities- what was the context for that, in school, leisure time, outside regular school, personel: school staaf, music and culture school etc. Be clearer here.

Formation dance- do you mean jazz or contemporary? Be as specific as possible.

Author Response

Response to Reviewer 2 Comments

Point 1: Interesting and important article that highlights the importance of arts in a school environment and that early exposure in a child's life is central to a person continuing to participate and consume artforms as an adult. It is also interesting to read about Hungary's art education in schools and how it relates to the cultural heritage that Hungary has in the Zoltán Kodály method and other pedagogies.

Response 1: Thank you.

Some improvements for suggestions:

In abstract: define "art" , is it artforms you mean?

Response 1: Yes, we examined a various artforms.

In text:

Point 2:. Define what you mean by "high and low culture" in a contemporary context. Bourdieu is not enough, needs to be further analysed with relevant updated references.

Response 2: Yes, you are right, the definition of high culture requires a separate approach, so it is not worth mentioning in the study. The theoretical framework of the study is cultural capital, so I have presented it in detail.

Point 3: Describe the parent group, how was the questionnaire distributed?

Response 3: We introduced the research, the sample and questionnaire in more detail.

Point 4: Write the age of the pupils in the study, not only the grade.

Response 4: OK, I wrote it.

Point 5. Define extracurricula artistic activities- what was the context for that, in school, leisure time, outside regular school, personel: school staaf, music and culture school etc. Be clearer here.

Response 5: Yes, thank you, I has introduced what do we mean about extracurricular artistic activities in more detail.

Point 6: Formation dance- do you mean jazz or contemporary? Be as specific as possible.

Response 6: Dance as an art form has been used as an umbrella term.